# Public’s Experience with an Online Reservation System for Residual COVID-19 Vaccines and the Potential for Increasing the Actual Vaccination Rate

**DOI:** 10.3390/vaccines11061021

**Published:** 2023-05-24

**Authors:** Seonah Lee, Suhyun Kim

**Affiliations:** 1College of Nursing, Chonnam National University, Gwangju 61469, Republic of Korea; 2Department of Nursing, Nambu University, Gwangju 62271, Republic of Korea

**Keywords:** COVID-19, public health, online reservation, residual vaccines, vaccination rate

## Abstract

This study aimed to investigate the public’s experience of online reservation of residual COVID-19 vaccines in an additional vaccination program. Online reservation was used to predict the vaccination rate. A sample of 620 participants completed the online survey between July and August 2021. About 38% of the participants made the online reservation. About 91% had a vaccination intention. Online reservations showed significant differences in their distribution according to age group, educational level, past flu shot experience, and COVID-19 vaccination intention. A negative experience was the most common response, which was mostly attributed to the difficulty in making an online reservation due to reservations being full. Positive experiences included updated information and notifications on the residual vaccines available, being able to choose a vaccination clinic, and the ease of making, changing, and canceling a reservation. About 72% reported the positive effect of residual vaccine usage on herd immunity. The results of this study suggest that when developing another online reservation program for vaccination, it is necessary to consider and address the negative experiences of the public with online reservations. The additional vaccinations may have resulted in an increased vaccination rate. Vaccination reservations can be used as an indicator to predict the actual vaccination rate and as a measure of a positive attitude toward COVID-19 vaccination.

## 1. Introduction

The COVID-19 pandemic, caused by the SARS-CoV-2 virus, has become a global health crisis since its emergence in December 2019, leading to millions of deaths worldwide. While vaccination has been recommended as one of the best ways to prevent and slow down the transmission of COVID-19, the vaccination rate varies across different regions and populations [1]. The World Health Organization (WHO) has emphasized the importance of vaccine equity and access, highlighting the need for countries to prioritize vaccinating the most vulnerable populations and to address vaccine hesitancy [2]. In South Korea, the government has launched an online reservation program for residual COVID-19 vaccines to increase vaccine accessibility and reduce vaccine waste. The program enables individuals to reserve leftover vaccines the same day through map platforms provided by Naver and Kakao, two of Korea’s top internet portals [3]. Effective delivery of information on vaccine availability can encourage more people to get vaccinated, particularly those who may have limited access to vaccination clinics or have difficulty scheduling appointments [4]. The online reservation program for residual COVID-19 vaccines was developed using Geographic Information Systems (GIS) and artificial intelligence (AI) technologies for the effective delivery of information on vaccine availability.

GIS and AI technologies have been increasingly used to promote good health and deliver effective healthcare [5]. In particular, GeoAI technologies can leverage spatial information and machine learning algorithms to analyze and visualize patterns and trends in health data, enabling decision-making and interventions at the population and individual levels [6]. The integration of GIS and AI can enhance vaccine accessibility and optimize the distribution of vaccines, particularly in resource-limited settings [7].

Several countries have implemented programs that combine online reservations and GIS for COVID-19 vaccination [8,9,10,11]. Israel has implemented a successful COVID-19 vaccination program that includes online reservations and GIS. The program used GIS to identify high-risk populations and prioritize vaccine distribution, while online reservation systems were used to schedule appointments and track vaccine administration [8]. The United States also implemented an online reservation and GIS-based COVID-19 vaccination program. Many states and local health departments have developed their online reservation systems, and GIS has been used to identify high-risk populations and target vaccine distribution [9]. These studies suggest that online reservation and GIS-based COVID-19 vaccination programs can be effective in increasing vaccination rates [8,9,10,11] and reducing COVID-19 cases and mortality rates [12,13,14,15].

Despite the potential benefits of an online reservation program and GeoAI technologies, little is known about the public’s experience with these programs. Previous studies have mainly examined vaccination intention as a factor predicting actual vaccination [12,13,14,15]. Many other factors, such as socioeconomic status, education level, geographic location, and cultural and social factors, can influence vaccination rates and attitudes [16,17,18,19].

However, online reservation systems, when combined with other data sources, such as GIS and AI, can provide other valuable insights into vaccination rates and attitudes [8,9,10,11]. These systems allow for the collection and analysis of the demographics of people being vaccinated, the vaccination rates by area, and the factors that influence people’s decision to get vaccinated. These insights can help health officials develop targeted vaccination campaigns and allocate resources more effectively. Therefore, online reservation systems, along with other data sources and tools, can contribute to predicting vaccination rates and attitudes toward COVID-19 vaccination. Above all, online reservation and GIS-based COVID-19 vaccination programs can be effective in increasing vaccination rates and reducing COVID-19 cases and mortality [12,13,14,15]. Despite these strengths, the public’s experience of the online reservation system of residual COVID-19 vaccines has not been explored.

Therefore, this study aimed to investigate the public’s experience of the online reservation of residual COVID-19 vaccines that was used as an additional program for vaccination.

## 2. Materials and Methods

### 2.1. Study Design

This study is a cross-sectional study using an online survey conducted between July and August 2021 in South Korea, when vaccination was expanded to all adults. Vaccination began in February 2021 for COVID-19-related medical staff, residents of long-term care facilities for the elderly, and workers and was expanded to the elderly over 65 and all medical staff in the second quarter and to other adults in the third quarter in South Korea [3,12].

### 2.2. Participants

The eligible participants for this study were adults over the age of 18 not infected with COVID-19 and consented to voluntary participation in an online survey. The exclusion criteria for study participants were those who had COVID-19 and were over 65 years of age.

### 2.3. Data Collection

Participants were recruited from a national opt-in panel of the SurveyBilly (Seoul, Republic of Korea) survey company. SurveyBilly maintains constant population proportions by age, population, and region on a national basis. The company used the research participation recruitment document provided by the authors, which included the title and purpose of this study, inclusion and exclusion criteria for participation, the data collection method, the time required to complete the online survey, anonymity, voluntary participation in the study, and a link to access an online informed consent form and a questionnaire created in Google Forms (Google, Mountain View, CA, USA). When a participant clicked on the link, they would first see an informed consent form and be asked to mark the consent checkbox to indicate their voluntary participation. The participants were then asked to complete a survey immediately after completing an informed consent form. Between July and August 2021, a total of 620 individuals completed the survey while the link to access the questionnaire was active. However, due to the remote nature of the questionnaire, it was challenging to determine the number of individuals who dropped out midway after initially attempting to participate in the study. The participants received an electronic coffee coupon worth $7 as a token of gratitude for completing the online survey.

### 2.4. Questionnaire

The questionnaire developed for this study included questions on vaccination intention (one item, yes or no), the completion of the online reservation (one item, yes or no), the online reservation experience (seven items, 5-point Likert scale), and the effect of residual vaccine usage on herd immunity (one item, 5-point Likert scale). Sociodemographic data included age, gender, educational level, religion, marital status, occupation, average monthly income, health status, the presence of chronic disease, flu shot experience in the past three years, and COVID-19 vaccination intention. Free text boxes were provided for the participants to enter advantages, disadvantages, and suggestions for the improvement of the online reservation system, the reasons for not making an online reservation, and the reasons the online reservation system is helpful for herd immunity. The questionnaire went through a validity test to confirm that there were no problems with readability and comprehension. The questions used in the questionnaire were evaluated for content validity by research experts. The questionnaire was finalized after any complicated and ambiguous sentences and difficult questions were revised. It took about ten minutes to respond to the questionnaire. 

### 2.5. Data Analysis

Using IBM SPSS Statistics 27.0 for Windows, categorical variables were analyzed using frequencies and percentages, and continuous variables were analyzed using means, standard deviations (SDs), frequencies, and percentages. Pearson’s chi-squared test was used to assess significant differences in the distribution of online reservations according to sociodemographic characteristics.

### 2.6. Ethical Consideration

This study was approved by the C University Bioethics Review Committee (IRB No. 1040198-210720-HR-050-03). The informed consent described in Google Forms explained that voluntary participation is important and that the survey can be stopped at any time during the questionnaire. The steps were set and classified so that only those who read the explanation and voluntarily agreed to participate in the study could respond to the questionnaire.

## 3. Results

Of the 620 participants that completed the online survey (100% response rate), 37.6% made an online reservation, and 91.3% intended to be vaccinated (Table 1). Online reservations showed significant differences in their distribution according to age group, educational level, past flu shot experience, and COVID-19 vaccination intention.

The online reservation experiences were similar when using a smartphone, PC, or both (Figure 1). The most common experience noted was difficulty in making the online reservation because the residual vaccine reservations ended early every time. This difficulty was also the drawback of the online reservation system that was most frequently noted in the free text boxes. Many participants reported that they had to click a mouse “at the speed of light.” However, the other online reservation experiences were positive based on mean scores over three out of five points (Figure 1). The participants also noted that an advantage of the online reservation system was to be able to make a reservation for vaccination anytime and anywhere. A disadvantage commonly noted was that the online reservation system was unsuitable for older people unfamiliar with working online. For improvement of the online reservation system, the participants suggested dividing people into ten groups by year of birth for more equal chances for reservations. They also suggested allowing more than five clinics per person to provide notifications for online reservations, securing a sufficient amount of vaccines, and limiting the use of macro software that carries out repetitive actions on a computer such as mouse movements, clicks, and keystrokes.

The participants also noted some reasons for not making an online reservation, where 26.4% of the 387j participants did not know how, 18.9% were already vaccinated, and 16% did not want to get vaccinated. However, 71.5% of the 620 participants reported that residual vaccine usage would be helpful for herd immunity, whereas 6.3% did not (mean ± SD = 3.84 ± 0.84). The main reasons the participants thought contributed to the online reservation system helping herd immunity were increased accessibility to vaccines (14.2%), an increased vaccination rate (13.1%), and immunity (12.6%). The top three reasons the participants thought the online reservation system might not be helpful for herd immunity were the possibility of a break-through infection due to a variant virus (2.7%), no decrease of confirmed cases (1.8%), and low confidence in vaccine effectiveness (0.8%).

## 4. Discussion

This study investigated the public’s experience of an online reservation system for residual COVID-19 vaccines. We found that there were significant differences in the distribution of online reservations according to age group, educational level, past flu shot experience, and COVID-19 vaccination intention. Similar results have also been reported in previous studies [16,17,18,19]. In addition to online reservation systems, governments can offer other options for individuals who do not have access to the internet or are not comfortable using online systems. These alternatives could include phone hotlines, walk-in clinics, or outreach programs to reach marginalized communities [20]. Moreover, support is needed to improve digital literacy for vulnerable populations. Governments can invest in digital literacy programs to help individuals who may not be comfortable using online systems to access vaccination appointments. These programs can help bridge the digital divide and ensure that vulnerable populations are not left out of the vaccination process [21,22,23,24].

The fact that there were significant differences in the distribution of online reservations according to COVID-19 vaccination intention is an interesting finding. The major shift in vaccination intent, where 47.6% of respondents who did not previously intend to get vaccinated changed their minds and now intend to get vaccinated, was investigated as a general characteristic of the study participants. However, the major shift in vaccination intent was not related to online reservations. Since the purpose of the study was not to investigate the reasons behind the major shift in vaccination intent, the general characteristics did not inquire about them. Furthermore, at the time of this study, the Korean government was sequentially distributing vaccines based on the degree of health vulnerability and actively promoting vaccination in addition to online reservation [3,12]. Therefore, the reasons and causes of the change will need to be investigated through other studies.

This study found that the public had more positive than negative experiences with making online reservations for residual COVID-19 vaccines via PC or mobile app. The most commonly noted negative experience was difficulty in making the online reservation because residual vaccine reservations were almost always full. The positive experiences noted by the participants were updated information and timely notifications on residual vaccines available, being able to choose a vaccination clinic, and the ease of making, changing, and canceling a reservation. Most participants intended to get vaccinated and reported the positive effect of residual vaccine usage on herd immunity. The most commonly reported reason that the online reservation system was thought to be helpful for herd immunity was the increased chances of vaccination against COVID-19. These results are similar to a previous study [15] that described the implementation of a GIS-based online reservation system for COVID-19 vaccination in Saudi Arabia. This previous study examined the effectiveness of an online reservation system in increasing vaccination coverage rates, the satisfaction of users, and the challenges and barriers faced during implementation. These problems also were shown in another study [14] that described the development and implementation of a GIS-based online reservation system for COVID-19 vaccination in India. This Indian study examined the features of the online reservation system, the impact of the system on vaccination coverage rates, and the challenges and barriers faced during implementation. Providing accurate and timely information about infectious diseases and vaccines and delivering information on available vaccines more efficiently are needed to improve the online vaccine reservation program [4,12,14,15].

Online reservation effectively increases vaccine accessibility by showing the nearest hospital offering vaccinations based on the user’s global positioning system (GPS), allowing the user to select the most convenient hospital [12]. These findings suggest that additional strategies should be explored to further increase vaccination accessibility by integrating AI and GIS [12,13,14,15]. Additionally, strategies for how we use GIS and AI technologies to increase vaccination coverage or intention should be explored. First, these strategies would be useful to identify populations in need [12,13,25,26]. For example, GIS can be used to identify populations that may be at risk for lower vaccination rates, such as those living in rural areas or areas with a lower socioeconomic status. AI can be used to analyze demographic data and health outcomes to identify correlations and potential areas of need. Second, it is necessary to develop targeted vaccination campaigns. GIS and AI can be used together to create targeted vaccination campaigns that focus on areas with lower coverage rates or populations with lower vaccination rates [25]. These campaigns can involve analyzing demographic data to identify which groups are most at risk and using GIS to identify the optimal locations for vaccination clinics [25]. Third, these strategies could be used for monitoring and tracking vaccination rates. GIS can be used to track vaccination rates across different geographic areas and can identify any changes or trends in coverage rates [12,13]. AI can be used to analyze these data and identify potential reasons for changes, such as the impact of vaccine misinformation or changes in vaccine availability [12,13]. Lastly, these strategies would be useful to predict vaccination demand. AI can be used to predict demand for vaccines based on various factors, such as demographic data, health trends, and historical vaccination rates [26]. These data can help health authorities plan for vaccine supply and distribution more effectively [26].

The online reservation system provides COVID-19 vaccines to people who were not grouped into the first waves of vulnerable populations, classified by criteria such as occupational cluster and underlying disease group [12]. The increased access and supply of vaccines through the online reservation system may have resulted in an increased vaccination rate because the online reservation system is an additional program for vaccination and facilitated the shift toward herd immunity. An additional study to identify this inference is required. As a large budget was required to develop and distribute the online reservation app, it is important to verify whether the app increased the vaccination rate [12,13].

Vaccination reservations can be a clear indicator to predict actual vaccination, along with vaccination intention, which existing studies mainly investigated [16,17,18,19]. Vaccination reservations can also be used as a measure of a positive attitude toward COVID-19 vaccination [10,11]. Age, educational level, past experiences with flu shots, and vaccination intention can be considered in planning a campaign to increase vaccination reservations [21].

Although this study provides valuable information, it has limitations. The results of this study cannot be generalized to the entire adult population of the country because the small sample only included those that were registered with SurveyBilly and may not be representative.

## 5. Conclusions

This study was conducted to explore the public’s experience of the online reservation system for residual COVID-19 vaccines, which was an additional vaccination program. Previously, vaccination intent was studied primarily in relation to vaccination prediction. However, this study showed that online reservations for vaccination could be used as an indicator to predict the actual vaccination rate and measure positive attitudes toward COVID-19 vaccination. This study suggests that healthcare practitioners and researchers can simultaneously consider several variables, including vaccination reservation and intention, actual vaccinations achieved, and attitudes toward COVID-19 vaccination. This approach will provide a comprehensive understanding to address vaccination issues in this pandemic.

## Figures and Tables

**Table 1 vaccines-11-01021-t001:** Participant characteristics and the distribution of online reservations for COVID-19 residual vaccines.

Characteristics	Total (*N* = 620)		Online Reservation		*p*
	*n*	%	Yes (*n* = 233, 37.6%)	No (*n* = 387, 62.4%)	
			% ^†^	% ^†^	
Age					
20 s	81	13.1%	39.5%	60.5%	0.002
30 s	243	39.2%	43.2%	56.8%	
40 s	193	31.1%	36.3%	63.7%	
50 s	67	10.8%	34.3%	65.7%	
60 s	36	5.8%	8.3%	91.7%	
Sex					
Male	284	45.8%	33.5%	66.5%	0.051
Female	336	54.2%	41.1%	58.9%	
Education					
<High school	110	17.7%	28.2%	71.8%	0.025
≥College graduate	510	82.3%	39.6%	60.4%	
Religion					
Yes	388	62.6%	36.1%	63.9%	0.319
No	232	37.4%	40.1%	59.9%	
Marital status					
No	272	43.9%	40.1%	59.9%	0.257
Yes	348	56.1%	35.6%	64.4%	
Occupation					
Employed	491	79.2%	38.5%	61.5%	0.360
Unemployed	129	20.8%	34.1%	65.9%	
Average monthly income					
<$2000	160	25.8%	34.4%	65.6%	0.208
$2000–$2999	166	26.8%	36.1%	63.9%	
$3000–$3999	168	27.1%	35.1%	64.9%	
$4000–$4999	70	11.3%	47.1%	52.9%	
≥$5000	56	9.0%	46.4%	53.6%	
Health status					
Healthy	319	51.5%	38.2%	61.8%	0.341
Average	257	41.5%	38.5%	61.5%	
Unhealthy	44	7.1%	27.3%	72.7%	
Chronic disease					
Yes	112	18.1%	36.6%	63.4%	0.814
No	508	81.9%	37.8%	62.2%	
Flu shot experience					
Yes	319	51.5%	45.8%	54.2%	<0.001
No	301	48.5%	28.9%	71.1%	
COVID-19 vaccination intention					
Yes	271	43.7%	42.4%	57.6%	0.002
Not in the past but yes currently	295	47.6%	36.9%	63.1%	
No	54	8.7%	16.7%	83.3%	

^†^ % per each row that sums to 100%, without missing values.

**Figure 1 vaccines-11-01021-f001:**
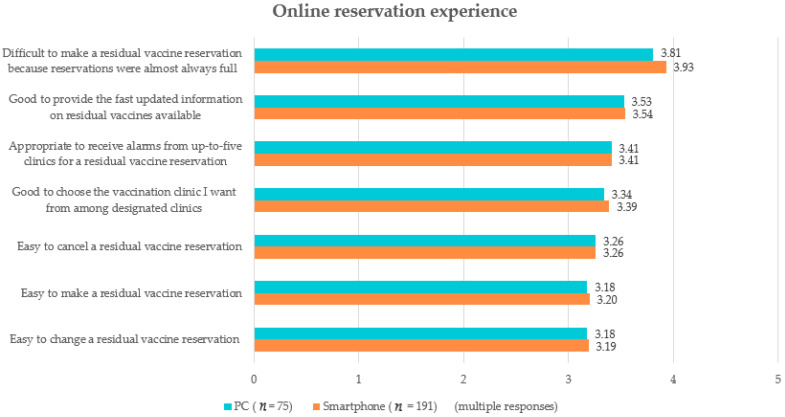
Experiences of the online reservation of residual COVID-19 vaccines.

## Data Availability

The data presented in this study are available on request from the corresponding author. The data are not publicly available due to ethical restrictions.

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
