# Peer review of "Public’s Experience with an Online Reservation System for Residual COVID-19 Vaccines and the Potential for Increasing the Actual Vaccination Rate"

_vaccines, 2023, doi:10.3390/vaccines11061021_

Round 1

Reviewer 1 Report

The abstract had no reference to the results of the chi-square analysis.

Introduction: The aim was to investigate the public's experience of online reservation of vaccines. Then, may I clarify what is the purpose of doing the chi-square analysis? Secondly, the authors stated that "Online reservation allows us to predict a vaccination rate and attitude toward COVID-19 vaccination." I think that is an over-generalization, because while it can predict vaccination rate, it cannot with accuracy elucidate attitudes towards COVID-19 vaccination.

I wonder why the need to state ethical approval twice in line 54 and line 89

Is the online reservation questionnaire valid and reliable? Would be good to provide Cronbach's alpha to indicate internal consistency reliability

In Table 1, kindly make the following amendments:

Religion --> had/didn't is not grammatical and is difficult to understand

Not sure if unmarried is a valid term

Average monthly income is in Won or USD?

COVID vaccination intention --> "had", "didn't" also needs to be rephrased

Results: There is no description of chisquare test results in the results section. It is a very important piece of information, in my opinion, because it highlights that those who were of lower education, older, and no flu shot experience are less likely to book vaccination online. 

Discussion: Thus, the results above needs to be discussed from the view of health equity if possible, and how should this be overcome, since online reservation seems to favour the more advantaged segments of the population and has the tendency to leave vulnerable populations out?

Conclusions: I do not agree that this survey can be used to understand the attitude towards COVID-19 vaccination, as there is no evidence or measurement to show that.

Reviewer 2 Report

The manuscript “Public’s Experience of the Online Reservation of COVID-19 Residual Vaccine and the Potential for Increasing an Actual Vaccination Rate” deals with a quite interesting topic, which is the influence of online reservation on the vaccination rate and attitude toward COVID-19 vaccination. Although the manuscript is adequately presented, several sections of the manuscript are confusing and misleading, in particular the introduction and the materials and methods section. Moreover, being a communication, according to the Journal directive, the manuscript should present groundbreaking preliminary results or significant findings that are part of a larger study over multiple years. I'm sorry to say that the article needs a lot of editing before it can be processed further.

Introduction

The introduction is rather confusingly articulated and the background of the study needs to be expanded. The sentence “hereafter referred to as the online reservation” is repeated three time in the text.

Materials and Methods

The study design is not described. The participants section contains many repetitions and the characteristics of the population examined are not adequately described. The authors should add information related to the construction and validation of the questionnaire.

Results

The results section is also extremely confusing. The response rate is rather predictable, considering that completing the questionnaire allowed to receive a financial compensation. However, it is not clear to me whether the possibility of making the online reservation was given at the same time as completing the questionnaire. If not, for how long was this possibility given? Was it given to the entire population (of which we have no information) or only to those who would then have filled out the questionnaire?

Discussions

In the discussion, the authors should make comparisons with other scientific articles existing in the literature. They should also explain how online reservations affected vaccination coverage rates.

References

Bibliographic references ought to be expanded.

Round 2

Reviewer 1 Report

Thanks very much for an opportunity to review this paper again, which I am glad looks much improved. There are a few more comments:

I was that that this questionnaire was with a response rate of 100%, Are the authors very sure that everyone they approached answered the questionnaire? Usually in remote questionnaires we are not able to determine the response rate. I would appreciate if the authors could clarify this.

In the discussion section, ll. 183-202, I'm wondering if this paragraph is suitable for the introduction section or the implications section at the end of the discussion section?

ll. 249-250 - "The increased vaccines may have resulted in an increased vaccination rate". I think it is not clear what the authors meant by "increased vaccines". Did the authors mean the increased access of vaccines through the online reservation system?

The authors mentioned in ll. 255.258 that "Some people have developed macros...". Is this one of the objectives of the research, to investigate if they have developed macros to hoard vaccines? If not, I suggest to leave it out of the discussion section.

OVerall comment: The discussion section should note the main results o the study (based on the study objectives), and discuss those results only. Then, afterwards, the authors may discuss the implications of the study, etc. I see that much of the discussion content seems unrelated to the aim of the study, and some discussions are too far fetched from the original research questions. I would like to suggest that the authors discuss what is relevant to this study only, and the implications should follow. Thank you

Reviewer 2 Report

I would like to thank you for the opportunity to review again the manuscript “Public’s Experience of the Online Reservation of COVID-19 Residual Vaccine and the Potential for Increasing an Actual Vaccination Rate”. Although I really appreciate the efforts made by the authors, unfortunately I have to admit that I cannot recommend this manuscript for publication.

The authors only partially and confusedly responded to my observations, and several sections of the manuscript are still confusing and misleading. The study population is not limited to the study sample and its characteristics, but comprise the characteristics of the entire population in which the study is conducted, also including the general level of education and the socio-economic characteristics. The construction and validation of the questionnaire is not treated in detail. The authors did not answer my questions regarding the results section and they did not explain how online reservations affected vaccination coverage rates.
